# Bacterial and Viral Co-Infection in the Intestine: Competition Scenario and Their Effect on Host Immunity

**DOI:** 10.3390/ijms23042311

**Published:** 2022-02-19

**Authors:** Siqi Lian, Jiaqi Liu, Yunping Wu, Pengpeng Xia, Guoqiang Zhu

**Affiliations:** 1College of Veterinary Medicine (Institute of Comparative Medicine), Yangzhou University, Yangzhou 225009, China; mx120180715@yzu.edu.cn (S.L.); mx120190706@yzu.edu.cn (J.L.); mx120190734@yzu.edu.cn (Y.W.); yzgqzhu@yzu.edu.cn (G.Z.); 2Jiangsu Co-Innovation Center for Prevention and Control of Important Animal Infectious Diseases and Zoonoses, Yangzhou 225009, China; 3Joint International Research Laboratory of Prevention and Control of Important Animal Infectious Diseases and Zoonotic Diseases of China, Yangzhou University, Yangzhou 225009, China; 4Joint International Research Laboratory of Agriculture and Agri-Product Safety of Ministry of Education of China, Yangzhou University, Yangzhou 225009, China

**Keywords:** bacterial–viral co-infection, signal transmission, neuroimmunity, nutritional immunity, intestinal microbiome

## Abstract

Bacteria and viruses are both important pathogens causing intestinal infections, and studies on their pathogenic mechanisms tend to focus on one pathogen alone. However, bacterial and viral co-infections occur frequently in clinical settings, and infection by one pathogen can affect the severity of infection by another pathogen, either directly or indirectly. The presence of synergistic or antagonistic effects of two pathogens in co-infection can affect disease progression to varying degrees. The triad of bacterial–viral–gut interactions involves multiple aspects of inflammatory and immune signaling, neuroimmunity, nutritional immunity, and the gut microbiome. In this review, we discussed the different scenarios triggered by different orders of bacterial and viral infections in the gut and summarized the possible mechanisms of synergy or antagonism involved in their co-infection. We also explored the regulatory mechanisms of bacterial–viral co-infection at the host intestinal immune interface from multiple perspectives.

## 1. Introduction

Diarrhea is one of the leading causes of morbidity and mortality in humans globally [1], which has a greater impact on children under 5 years of age [2,3,4]. In addition to human disease, various poultry and domestic animals lose their economic value due to diarrhea [5,6,7,8]. Gastrointestinal infections can be caused by a variety of pathogens, such as bacteria, viruses, and parasites. Possible causative pathogens include rotavirus (RV) A, norovirus (NoV) GI and GII, adenovirus, *Salmonella*, *Campylobacter jejuni*, *Shigella*, and *Escherichia coli* (*E. coli*) [2,9]. Co-infections of these pathogens frequently cause more severe consequences than single-pathogen infection [10]. Although the mechanisms of single infections by these pathogens have been studied considerably, little is known about the regulatory mechanisms of co-infections between them [11]. Diarrhea caused by bacterial and viral co-infection is an important health problem in developing countries [12]. Understanding the mechanisms of co-infection is essential for the development of more precise disease control strategies.

Arpit Kumar Shrivastava et al. performed pathogenic tests on stool samples of 130 children with acute diarrhea from a pediatric clinic in India and showed that the pathogenic organisms were mainly *E. coli* (30.07%), followed by RV (26.15%), *Shigella* (23.84%), adenovirus (4.61%), *Cryptosporidium* (3.07%), and *Giardia flagellates* (0.77%), and 44 of 130 cases (33.84%) were infected with two or more pathogens simultaneously [4]. Shilu Mathew et al. tested the pathogens of 70 pediatric patients with gastroenteritis, and there were varying degrees of mixed viral and bacterial infections, including RV, NoV, Enteroaggregative *E. coli* (EAEC), and Enteropathogenic *E. coli* (EPEC). Compared to RV and NoV infections alone, EPEC and EAEC co-infections aggravate the frequency of diarrhea and vomiting [13]. Co-infections lead to altered intestinal flora abundance, decreased intestinal microbial diversity, and increased intestinal flora disorders [14,15], such as accomplished with a higher abundance of Clostridiaceae and Streptococcaceae within the gut, the microbiota can interact directly with epithelial cells, leading to a series of events in the process of inflammatory [13]. In contrast, Sabrina J. Moyo et al. analyzed stool samples from 723 children with diarrhea in Tanzania and found that co-infections frequently occur in diarrhea, but that the pathogenicity of one pathogen is sometimes enhanced by co-infection and diminished by others, with no effect of co-infection on the severity of diarrhea [16]. This is different from the commonly understood perception that the more pathogens, the more severe the infection, suggesting that there are some as yet unexplained regulatory mechanisms in co-infection.

The gut harbors the largest mucosal surfaces and acts as the largest immune organ of the human body. The gut is exposed to a complex environment in which it is continuously exposed to stimuli from multiple foreign antigens [17,18]. The innate or non-specific immune system in the gut is our body’s first line of defense and consists of immune cells and other immune effectors. Innate immunity cells comprise a diverse group of cell types, including intestinal epithelial cells (IECs) [19], mononuclear macrophages, dendritic cells, natural killer cells, mast cells, granulocytes, M cells, etc. The immune effectors are mainly substances secreted by immune cells, such as mucin, secretory immunoglobulin A (sIgA), and defensins [20]. IECs, mononuclear phagocytes, and gut-associated lymphoid tissue play an important role in sensing elements of the gut microbiome and in regulating the immune response [21]. In addition to this, there is growing evidence that autonomic nerves and neurotransmitters, as well as neuropeptides, modulate the intestinal immune system and thus the intestinal inflammatory process [22]. The enteric nervous system (ENS) has widespread innervation to other organs of the body, including the central nervous system, by translating chemical signals from the environment into neuronal impulses to control and integrate all body activities. It forms an intricate network of enteric neuroimmunity to sense and respond to the dynamic ecosystem of the gastrointestinal tract [23]. Upon stimulation, the gut detects (or recognizes) the pathogen-associated molecular pattern molecules (PAMPs) and damage-associated molecular pattern molecules (DAMPs) through some different mechanisms [24], regulating downstream gene transcription and inflammatory responses [25].

Recent studies of viral–bacterial interactions demonstrated that eukaryotic virus–bacterial interactions may be pervasive and have serious implications for microbial pathogenesis, warranting further investigation [10]. Interactions between co-infecting pathogens within a host can alter pathogen transmission, and their effects generally depend on the relative order of arrival of pathogens within the host (intra-host preference effect) [26]. The clinical and basic study of bacterial and viral co-infection in the respiratory epithelium has been repeatedly highlighted. There are more reported clinical cases of co-infection in the intestinal epithelium, but the study of interaction mechanisms remains in the initial stage [27]. In this review, we discussed the different scenarios of bacterial and viral co-infection in the intestine and the impact of co-infection on host intestinal immunity, and also summarized the currently known molecular mechanisms and present salient issues in the field of co-infection with intestinal pathogens that may provide new ideas for future research.

## 2. The Real “Killers” Are Bacteria or Viruses? The Order of Infection Is Important

The initial research on bacterial–viral co-infection commenced with the “Spanish Flu”, where 95% of the mortality was attributed to co-infection by *Streptococcus pneumoniae* and *Staphylococcus aureus*. Earlier researchers believed that this was a comparatively straightforward opportunistic approach in which naturally dormant bacteria, such as *Staphylococcus aureus*, took advantage of the reduced immunity caused by primary viral infections to become pathogenic. However, this is over-simplistic; while co-infection often enhances and prolongs symptoms, it can alleviate them in some cases [28]. Although this condition is found in co-infections with respiratory pathogens, it is also present with a similar situation in co-infections in the gut.

### 2.1. Viral Infection Followed by Bacteria/Bacterial Threaten

When host intestinal cells are invaded by viruses, the state of cell membranes disruption and immune function suppression leads to increased vulnerability towards bacterial infection in the intestine; however, when the intestine is less susceptible to infection by pathogenic bacteria and in a defensive state, certain viruses can stimulate the immune system of the intestine. In this section, we describe the different effects of secondary bacterial infections following enteroviral infections.

#### 2.1.1. The Virus Infects First and Promotes Bacterial Infection

Cellular modifications induced by viruses that naturally inhabit the gut may modulate the ability of other pathogens (e.g., bacteria or viruses) to adhere to and invade epithelial cells and may produce or favor more severe infections in humans. The ways in which viruses synergize bacterial infection include individual or combined mechanisms, such as upregulation of cellular receptors, disruption of epithelial barriers, and suppression of the immune system [29,30].

Transmissible gastroenteritis virus (TGEV) is a coronavirus characterized by diarrhea and high morbidity, and persistent TGEV infection induces intestinal epithelial–mesenchymal transition (EMT), i.e., cellular conversion from epithelial cells to mesenchymal cells promoted motility and invasiveness and enhanced the adhesion of *Enterococcus faecalis* (*E. faecalis*) and enterotoxigenic *E. coli* (ETEC) K88 to cells [31]. The pathogenic bacteria first bind to the bacterial receptor fibronectin through the Zipper mechanism and further bind to the membrane protein integrin 5 to invade the cell. It also invades the host through the enlarged cell gap. Reversal of EMT downregulates the expression of integrin 5 and fibronectin and inhibits the adhesion of ETEC K88 to IECs (Figure 1A) [7,31].

RVs are double-stranded RNA viruses belonging to the family Reoviridae and infect the small intestine via the oral route. In a human colon adenocarcinoma cell model, RV infection makes Caco-2 cells more susceptible to *Yersinia pestis* and *Yersinia pseudotuberculosis* in the small intestine by strengthening early bacterial–cell interactions and internalization. The intracellular proliferation of bacteria leads to intracytoplasmic organelle damage, resulting in reduced viral antigen synthesis in the midst of co-infection [32]. *Aeromonas* is a group of Gram-negative short rods of the family Vibrionaceae, which are important pathogens of fish and occasionally cause diarrhea and extraintestinal infections in humans and animals [33]. Pre-infection of Caco-2 cells with RV can also affect the adherence properties of some *Aeromonas* species to host cells, and this effect varies with the duration of viral infection (Figure 1A) [34].

*Listeria monocytogenes* (*L. monocytogenes*) is a Gram-positive bacterium, and a key step in its pathogenesis is the invasion of this microorganism into IECs [35]. The co-infection of RV or poliovirus (PV) and *L. monocytogenes* were investigated in Caco-2 cells, where RV-infected cells exhibited an enhanced internalization of *L. monocytogenes* and promoted bacterial replication, whereas PV infection had only a slight disruptive effect on bacterial entry before infection, hindering bacterial proliferation to some extent (Figure 1A) [36].

#### 2.1.2. The Virus Infects First and Inhibits Bacterial Infection

Upon viral infections, the host’s innate immune system responds to the stimulation of the pathogen immediately; it can limit the pathogenic infection and subsequently trigger appropriate signals to activate an adaptive immune response to clear the infecting microorganism. In contrast, when the immunosuppressive class of viruses first colonizes the gut, it suppresses the host immune system and leads to easier colonization of the bacteria [37]. For example, enterovirus 71, which can initially colonize the intestine and cause systemic lesions, activates the immune system and suppresses infection by other pathogens at a later stage [37,38].

Barton et al. reported that mice latently infected with mouse gamma-herpesvirus 68 or mouse cytomegalovirus had inhibitory effects on *L. monocytogenes* [39]. The inhibitory effect is dependent on the systemic activation of macrophages and the production of cytokines in the underlying innate immune system, including the production of the cytokine interferon-γ. Based on these results, it can be hypothesized that latent infection by viruses adjusts the host to an immunity mode and prepares the host to fight against the challenge of bacteria [39,40]. Astroviruses can also mimic the role of the microbiota to set the immune threshold and protect the epithelium from intestinal pathogens by inducing type III IFN expression in IECs [41]. NoV infection can play a protective role in DSS-induced intestinal injury and murine *Citrobacter*-induced intestinal infections via inducing an IFN-I (α/β) response in the mouse colon and promoting Interleukin 22 (IL-22) production by natural lymphocytes (Figure 1B) [42].

#### 2.1.3. The Virus Infects First, and Its Replication Is Affected by the Later Infecting Bacteria

Interestingly, the determinants of infection are not in accordance with the infected order of microorganisms. In some cases, it is not only the former that affects the latter in co-infection, but the latter can also act on the one that colonizes first. As early as 1996, F. Superti et al. found that co-incubation with *L. monocytogene**s* at 37 °C for 5 h leads to an increase in RV antigen synthesis, but it did not affect PV replication in Caco-2 cells [36]. This article demonstrated that infection of enterocyte-like cells by different enteroviruses could affect the susceptibility of cells to secondary bacterial invasion, but the mechanism was not explored in depth.

NoV and *Salmonella* are major pathogens that affect human public health security. Murine NoV (MNV) activates caspase-mediated apoptotic pathways that kill host cells and establish productive infections [43]. *Salmonella* is an invasive pathogen that colonizes host cells, preventing cell death and enhancing cell survival, thereby establishing long-term infection within host cells [44]. Infection of RAW 264.7 cells by mouse NoV causes apoptosis and increased replication of *Salmonella* Heidelberg through cleavage of poly ADP ribose polymerase (PARP), caspases 3 and 9 [43,45]. Nevertheless, infection with *Salmonella* Heidelberg blocked the apoptotic process caused by MNV, and this might be related to the activation of the PI3K/Akt pathway induced by the type III secretion system effector SopB (Figure 1C) [45].

### 2.2. Bacterial Infections Followed by Viruses

Inter-boundary interactions between bacteria and viruses play a crucial role at the host–pathogen interface. Similar to 2.1, bacterial infections have two effects on subsequent viral infections. On the one hand, enteric bacteria may also assist a wide range of enteroviruses in infections, e.g., many enteric bacteria improve PV miscibility efficiency and fitness by promoting viral DNA recombination [46]; certain polysaccharides and lipids on the surface of enteric bacteria can improve the thermal stability of some enteric viruses [10]. On the other hand, enteric bacteria can inhibit viral infection, e.g., *Salmonella* inhibits NoV replication in mouse macrophages [45]; the bacteriocin CRL35 (ECRL) secreted by *E. faecalis* inhibits the synthesis of viral glycoproteins necessary for viral infection and replication [47]; the presence of microbacteriocins with antiviral activity in bacterial supernatants significantly reduces viral titers in vitro [48].

#### 2.2.1. Bacteria Infect First and Promote Viral Infection

The microbiota enhances the replication and spread of enteroviruses through several mechanisms. We commonly understand that bacteria can suppress the innate immune response to protect viruses from clearance [46,49]. The direct binding of bacterial surface polysaccharides can enhance the stability of enteric virions and increase viral attachment to host receptors [50].

Certain strains increase viral co-infection of mammalian cells. Bacteria-mediated viral co-infection is associated with bacterial adherence to cells [50]. RNA viruses such as PV, enteroviruses, and NoVs exist as genetically diverse groups of viruses with different adaptations, and mutations in their RNAs during replication may lead to deleterious outcomes [51]. It was found that enteric PV bind directly to a variety of bacteria, thus promoting stronger viral infection of host cells and mediating more efficient viral recombination. Importantly, bacterial strains involved with viruses in causing co-infections contribute to promoting genetic recombination between two different viruses, thereby eliminating deleterious mutations, restoring viral fitness, and producing progeny with the ability to grow under otherwise limiting conditions (Figure 1D) [46].

Enteroviruses can bind to bacteria via bacterial surface polysaccharides. Tissue blood group antigens (HBGAs) expressed on the intestinal epithelium are thought to be receptors or co-receptors for human NoVs (HuNoVs) [52]. HuNoVs interact with HBGAs glycan and bind to specific bacteria [53]. The binding of enteroviruses to bacterial envelope components also enhances their thermal stability (Figure 1D) [54]. Direct binding of bacterial surface polysaccharides enhances the stability of enterovirus particles and increases the attachment of the virus to host receptors.

Sharon K. Kuss et al. found that the viability of PV was significantly increased after incubation with Gram-negative (*E. coli*) or Gram-positive (*Bacillus cereus* (*B. cereus*), *E. faecalis*) bacteria. Exposure to *B. cereus* resulted in increased adhesion of PV to Hela cells and increased infectivity by more than 500% [55]. Further studies revealed that bacterial components-lipopolysaccharide (LPS), peptidoglycan (PG), and other N-acetylglucosamine-containing polysaccharides increased PV bindings to its receptors and viral shedding (Figure 1D) [55]. It follows that binding to bacterial components increases the stability of the viral shell exposed to high temperatures and promotes environmental adaptation of enteroviruses [56].

#### 2.2.2. Bacteria Inhibit Viral Infections

During the process of NoV and *Salmonella* co-infections, it was found that *Salmonella* enterica pre-infection of RAW 264.7 cells reduced NoV replication in mice by blocking viral binding to macrophages early in the viral life cycle and inducing the production of antiviral cytokines such as IL-6, IFN-γ, TNF-α later [11,45,57].

In addition to live bacteria, certain bacterial components were shown to induce innate antiviral immunity via Toll-like receptors (TLRs). Type 1 pilus adhesin FimH-induced innate antiviral immunity is associated with IFN-β production and requires the involvement of MyD88, Trif, TLR4, IRF-3, and type I IFN signaling (Figure 1C) [58]. Bacterial flagellin, a major component of the bacterial flagellum, can effectively activate host defense gene expression in IECs, which is considered a major immune activator in the gut and used as an adjuvant in the development of several vaccines [59,60]. Zhang et al. reported that systemic treatment of a mouse model with bacterial flagellin could prevent and cure RV infection. This protection can be extended to other enterovirus infections, including eutherian virus [61]. The induction of antiviral infection by flagellin is dependent on the activation of Toll-like receptor 5 (TLR-5) and NOD-like receptor C4 (NLRC-4) on the dendritic cell surface and the inductions of downstream cytokines IL-22 and IL-18 [62]. The combination of IL-18-induced apoptosis and IL-22-induced proliferation results in faster turnover of villi epithelial cells than the rate of viral infection of new cells (Figure 1C) [61].

### 2.3. Mechanisms of Synergism or Antagonism between Enteric Pathogens

It is becoming increasingly apparent that interkingdom communication and interactions between bacteria and viruses play critical roles at the host–pathogen interface [50]. In general, the effect of co-infection with enteric pathogens is divided into two major aspects: first, direct interaction between pathogens, such as the binding of bacterial surface polysaccharides to viral particles, enhances their thermal stability [54,56]. Second, indirect action, the host intestine is affected by pathogen infection and entered a “disrupted” state susceptible to infection by other pathogens, such as a damaged intestinal barrier, intestinal flora disorders, and upregulation of cell surface receptor expression [63], etc. (Figure 1E).

Antagonism between pathogens also exists, including, but not limited to: a nutritional competition between different pathogens [64]; secretions of certain bacteria inhibit the synthesis of essential viral proteins; the first infected pathogen activates an innate immune defense against later pathogens [39,40]; exhibits an immune defense state or stimulates the host to release a substance that is not harmful to itself but inhibits the infection of the latter, such as interferons, bacteriocins, antimicrobial peptides with antiviral activity, etc. (Figure 1E) [29,57].

## 3. Impact of Pathogen Interactions on Host Intestinal Immunity (A Masterful Encounter between Pathogen and Intestine)

The gut is a large and ordered whole; during infection, this balance can become disturbed. The link between bacteria, viruses, and intestinal immunity is intricate and multidirectional, and the interaction between these three involves different mechanisms [65]. This review broadly classifies these interactions and describes the details in five aspects: intestinal cells and their secreted effectors, signaling, neuroimmunity, nutritional immunity, and the gut microbiome.

### 3.1. Intestinal Immune Cells and Their Secreted Substances

The intestinal epithelium consists of six functionally defined differentiated cells: enterocytes, enteroendocrine cells, Paneth cells, tuft cells, cup cells, and microfold (M) cells [25]. These cells play roles in nutrient absorption, hormone secretion, antimicrobial peptide (AMP) secretion, gustatory chemosensory response, mucus production, and antigen sampling, respectively [66]. Although epithelial cells are not usually considered immune cells, they are highly responsive to infection and play a fundamental role in establishing immune defenses in the gut using both autonomous and involuntary mechanisms [67]. The tight cell-to-cell junctions constitute the intestinal barrier to control the entry of the microbiota, nutrients, and other substances from the lumen into the organism [18,68]. Certain proteins are present on the surface of enterocytes that can act as receptors for pathogens [68,69]. By stimulating the pathogen, the enterocytes produce antimicrobial substances, for example, inhibit local infection through secreted factors. In addition, enterocytes produce cytokines and chemokines that coordinate immune responses and induce systemic immunity through tissue-resident immune cells [67].

Viruses and intracellular bacterial pathogens (IBPs) invade monocytes/macrophages (MOs/MPs) and dendritic cells, epithelial cells, fibroblasts, and endothelial cells of the intestine. However, these ultimately differentiated cells remain in a quiescent metabolic state during infection and cannot accommodate the efficient intracellular replication of viruses and IBPs. Viruses and IBPs can reprogram host cell metabolism in pathogen-specific ways to increase the supply of nutrients, energy, and metabolites that require the pathogen to permit replication [70]. In short, most viruses manipulate the host cell metabolism through “pro-viral metabolic changes” to optimize the biosynthetic requirements of the virus. Besides, host cells have developed metabolic strategies to inhibit viral replication through “antiviral metabolic changes” [70,71].

Tuft cells are chemosensory cells associated with intestinal immunity through the production of cytokines such as IL-25 [72]. CD300lf is an MNV receptor. Cluster cells were found to be a rare type of IECs that express CD300lf and are the target cells of mouse intestinal MNV. Type 2 cytokines that induce the proliferation of cluster cells were found to promote MNV infection in vivo. These cytokines can substitute for the role of commensal flora in promoting viral infection [73].

Epithelial microfold (M) cells are located in the follicle-associated epithelium and can sample various substances, such as soluble antigens and microorganisms, through fluid-phase cytosolic drinking and receptor-mediated endocytosis [74]. M cell-dependent antigen uptake is mediated, at least in part, by specific receptors, such as β1-integrins, cellular prion proteins, and glycoprotein-2 (GP2). GP2-dependent bacterial uptake initiates antigen-specific secretion of M cells and can maintain intestinal immune homeostasis by mediating cellular immunity [75]. M cell-dependent antigen transport promotes a cellular immune response against intestinal commensal flora to alleviate pathogenic bacterial infection-induced colitis in mice [76]. However, various intestinal pathogens, such as *Shigella*, *Yersinia pestis*, Brucella abortus, eutherian virus, and itchy prion protein, utilize M cells as a portal for initial invasion [74]. Accordingly, M cell-dependent antigen uptake has both beneficial and detrimental roles in the milieu of mucosal infection and host defense (Figure 2).

Paneth cells are specialized IECs that limit bacterial invasion by secreting antimicrobial proteins, including lysozyme [77,78]. Pathogenic microorganisms can disrupt the Golgi apparatus, trigger endoplasmic reticulum (ER) stress, interfere with protein secretion and inhibit antimicrobial protein delivery, thereby inhibiting antimicrobial defense in the intestine. During bacterial infection, lysozyme can be secreted via secretory autophagy (an alternative autophagy-based secretory pathway), thus bypassing the traditional secretory pathway; bacterial-induced endoplasmic reticulum stress triggers secretory autophagy in Paneth cells, requiring exogenous signals from native immune cells [77]. AMPs are a large class of small, usually positively charged polypeptides that are against the cell membranes of bacteria, fungi, and other microorganisms [67]. Mammalian Paneth cells are the main AMP producers in the intestine. *Salmonella* infection causes an increased abundance of Paneth cells and IECs and extensively activates the antimicrobial program [79]. Paneth cells may indirectly influence the gut microbiota or the outcome of viral infection by regulating bacterial populations. Paneth cells also play a key role in host defense by sensing microbes through TLRs [78]. Thus, Paneth cells play an integral role in the formation of the microbiome and host defense.

Type III natural lymphocytes (ILC3s) are present in lymphoid organs and the intestine and are required for resistance to intestinal bacterial infections. G protein-coupled receptor 183 (GPR183) is a chemotactic receptor expressed on ILC3 in mice and humans. In vivo GPR183 deficiency results in disorganized distribution of ILC3 in mesenteric lymph nodes and reduces ILC3 accumulation in the intestine, and thus GPR183-deficient mice are more susceptible to infection by pathogenic intestinal bacteria [80]. In addition, ILC3 expresses vasoactive intestinal peptide (VIP) receptor 2 (VPAC2), and the VIP was identified to enhance resistance to murine *Citrobacter* infection by promoting CCR9 expression of ILC3 and intestinal recruitment [81].

*Salmonella* Typhimurium (STm) is an intestinal pathogen that triggers inflammation, and then inflammation reprograms the metabolic and immune environment of the intestine, thereby promoting STm expansion [82]. The intestinal mucosa secretes a series of antimicrobial peptides into the intestinal lumen, including the antimicrobial lectin RegIIIβ [83] and the novel bactericidal protein SPRR2A (small proline-rich protein 2A) [84]. RegIIIβ enhances mucosal inflammation by increasing the levels of TNF-α and IL-6 cytokines, Kc and Mpi2 chemokines, and lipocalin-2, in murine. The acute phase of STm infection promoted sustained intestinal colonization of STm [83]. RegIIIβ also profoundly altered the composition of the commensal flora during infection, decreasing the proportion of anaphylactic bacilli. This makes it prolong the persistent colonization of pathogenic bacteria as well as the duration of enteropathy during infections, while supplementation with specific mimics or vitamin B6 accelerates the clearance of intestinal pathogens and alleviates enteropathy [83]. The novel bactericidal protein SPRR2A (small proline-rich protein 2A) differs from the known flora-induced AMP in its phylogeny and mechanism of action, is induced by anti-parasitic type 2 immunity, and selectively inhibits Gram-positive commensal and pathogenic bacteria, protecting the intestinal barrier against bacterial invasion during intestinal helminth infections [84].

Interferons are produced by lymphocytes. Type I (IFN-α/β) and type III (IFN-λ1-4) IFNs exert synergistic antiviral effects on cells by inducing the transcription of hundreds of antiviral IFN-stimulated genes (ISGs) and promoting viral clearance [85]. There is substantial evidence that high levels of expression of type I IFN can impair antimicrobial immune responses, and in viral–bacterial co-infections, type I IFN is thought to be important in promoting secondary bacterial infections. There are different mechanisms known to be associated with type I IFN, including inhibition of Th17 and neutrophil responses; reduced neutrophil production, recruitment and survival; inhibition of neutrophil chemotactic agents; and reduced IL-17 production by T cells [86]. Type III IFNs (IFN-λ) are at the center of the storm of infection [85]. Type III IFNs protect adult and lactating mice from enterovirus infection and enterovirus-induced death [87]. Mouse RV induces IFN-l expression and IL-1α production in IEC, while IEC-derived IL-1α activates IL-22 expression in the intestinal ILC3 population from the lamina propria. IL-22 synergistically activates STAT1 with IFN-l to reduce enteric RV infection. Combined treatment with IL-22 and IL-18 protected mice from RV infection and pathological injury [67]. NoV infection can play a protective role in DSS-induced intestinal injury and murine *Citrobacter*-induced intestinal infection by eliciting an IFN-I response in the mouse colon to promote IL-22 production by natural lymphocytes (Figure 2) [42]. IFN-α and IFN-β prevented systemic infection by NoV in mice, but only IFN-λ controlled persistent intestinal infection, suggesting that IFN-λ specifically inhibits intestinal NoV [88]. Thus, IFN-λ plays an important role in protecting the intestinal mucosa from viral infection [57]. In addition, IFN-λ plays a role in a variety of intracellular and extracellular bacterial infectious diseases, including *L. monocytogenes*, *Streptococcus pneumoniae*, *Haemophilus influenzae*, *Staphylococcus aureus*, *Salmonella enterica*, *Shigella*, and *Mycobacterium tuberculosis* [89]. For example, infection of mouse placenta by *Listeria monocytogenes* resulted in upregulation of IFN-λ2/λ3 mRNA expression levels in the placenta, indicating that IFN-III contributes to epithelial protection from bacterial infection (Figure 2) [90].

The vitamin D receptor (VDR) belongs to a family of nuclear receptors that are highly expressed in the small intestine and colon and perform critical roles in local and systemic immunity, host defense, and host-microbe interactions [91]. VD-VDR signaling maintains intestinal barrier function through various mechanisms, such as upregulation of tight junction protein expression, inhibition of IEC apoptosis, promotion of autophagy in IEC, and enhancement of mucosal repair [91]. VDR has regulatory effects on both bacterial and viral intestinal infections. Rong Lu et al. found that lysozyme was significantly reduced in Paneth cells from Paneth cell-specific VDR knockout (VDR∆PC) mice, with diminished pathogenic bacterial growth inhibition and reduced autophagic response; VDR∆PC mice had enhanced inflammation following *Salmonella* infection, and the impairment of their autophagic response leads to a weakened defense against pathogenic bacterial invasion and promotes intestinal inflammation in mice [92]. By constructing knockout mice with specific deletion of VDR in IECs (VDRΔIEC), Paneth cells (VDRΔPC) and myeloid cells (VDRΔLyz), Jilei Zhang et al. analyzed the bacterial and viral composition and metabolites of each group of mice and found that tissue-specific deletion of VDR altered the viral population and functionally changed the viral receptor, which resulted in ecological dysregulation, metabolic dysfunction, and infection risk. Expression of TLR3/7, NOD1/2, and NLR6 was upregulated in knockout mice, and C-type lectin receptor 4L (CLR4L) was significantly upregulated (Figure 2) [93]. Overall, the significant alterations in pattern recognition receptors (PRRs) of conditional VDR knockout mice indicate an impact of VDR on intestinal homeostasis and PRRS expression. Lack of VDR in Paneth cells can lead to viral dysregulation, which may further lead to impaired epithelial barrier function. VDR activation has a regulatory role in enterovirus–host interactions [91,92,93]. This suggests that VDR could be a potential target for study in bacterial–viral–host interactions.

### 3.2. Immune and Inflammatory Signaling Pathways

The innate immune system coordinates a wide range of germline-encoded PRRs to sense PAMPs and DAMPs [24,94]. PRRs are mainly composed of TLRs, C-type lectin receptors, cytoplasmic DNA sensors (i.e., cyclic GMP-AMP synthases), and several other cytoplasmic receptors, such as RIG-I-like receptors or Nod-like receptors [24,95]. Typical inflammatory vesicles are cytoplasmic multiprotein complexes involved in intrinsic immunity, crucial for host defense and repair [96]. They are mainly assembled with nucleotide-binding domains and leucine-rich repeats (NLR) proteins, such as NLRP1, NLRP3, NAIP, and NLRP6 [62,97]. Upon activation, these sensor proteins recruit pre-caspase-1 directly or indirectly via adapters ASC or NLRC4, leading to caspase-1 dimerization and activation. Subsequently, caspase-mediated processing of pro-interleukin (IL)-1β, pro-IL-18, and the pore-forming protein gasdermin D (GSDMD) lead to maturation and release of these cytokines, as well as cellular scorching (Figure 2) [97]. There have been many studies of pathogen interference with host signaling pathways in respiratory co-infection, such as activation of the TLR2-MYD88-NLRP3 signaling axis mediating an increase in IL-1β during co-infection with influenza A virus and *Streptococcus pneumoniae* [98], and a similar mechanism exists in the intestine.

During intestinal pathogenic bacterial infection, innate immune cells in mice are activated and produce IL-23 and IL-22 to promote antimicrobial peptide production and bacterial clearance. IL-36R signaling promotes IL-23/IL-22/antimicrobial peptide and IL-6/IL-22/antimicrobial peptide-mediated inhibition of intestinal pathogenic bacterial infection by integrating innate and adaptive immune responses (Figure 2) [99].

TLR3 is present in IECs, and it recognizes viral infection and induces NF-κB signaling and IL-15 production [67]. RV genomic dsRNA and its synthetic analog polyinosinic-polycytidylic acid (poly (I: C)) cause severe mucosal damage in the small intestine. Upon binding to TLR3, dsRNA triggers the secretion of IL-15 by IECs, which disrupts mucosal homeostasis by acting on CD8αα + IELs [100]. Intestinal epithelial TLR3 expression correlates with RV susceptibility. Viral shedding was significantly increased in TLR3 or Trif-deficient mice compared to controls in RV infection experiments, and expression of proinflammatory and antiviral genes was reduced (Figure 2) [101]. TLR3 signaling exerts a dual effect on intestinal infection, contributing not only to host defense but also to viral pathogenesis. TLR4 is expressed on the surface of various phagocytic cells, including macrophages, peripheral blood monocytes, neutrophils, and DCs [24], either type I “hair” FimH [58] or LPS produced by Gram-negative bacteria can stimulate TLR4 and mediate the downstream NF-κB and MAPK signaling pathways (Figure 2) [102]. TLR5 recognizes extracellular flagellin and uses MyD88 to initiate MAPK signaling and NF-κB activation to stimulate cytokine secretion and trigger an inflammatory response to eliminate pathogens [103]. The flagellin of F4 ETEC induces TLR5-mediated IL-17C expression in IECs and increases the expression of antimicrobial peptides and tight junction proteins in an autocrine/paracrine manner that promotes mucosal host defense against bacterial infection [24,104]. Additionally, flagellin activates IL-22 and IL-18 activation through TLR5/NLRC4 (Figure 2) [62], driving IL-18-induced apoptosis of viral infection as well as IL-22-induced proliferation [61], thereby inhibiting viral infection.

It was found that epithelial NAIP/NLRC4 inflammatory vesicles drive heterogeneous cell death (including caspase-1-mediated scorch death and caspase-8/-3 mediated apoptosis) of IECs to reduce the tissue load of STm, to promote intestinal epithelial cell expulsion can reduce STm load by driving intestinal epithelial cell scorch death or apoptosis and promoting intestinal epithelial cell expulsion, thereby alleviating TNF-induced intestinal epithelial barrier disruption (Figure 2). NAIP/NLRC4 deficiency leads to increased TNF levels, intestinal epithelial barrier disruption, and impaired tissue regeneration in mice [105].

NLRP6 inflammatory vesicles are involved in the regulation of mucus secretion, antimicrobial peptide production, NF-κB, MAPK, and IFN signaling pathway and play an important role in host defense [106]. NLRP6 acts roles in different cells. In IECs, NLRP6 senses microbial-associated metabolites, forms ASC-dependent inflammatory vesicles, and promotes downstream IL-18 release and secretion of antimicrobial peptides (Figure 2) [96]. In the intestine, NLRP6 regulates IL-18 production, cupped cell function, and flora homeostasis and collaborates with NLRP9 in defense against viruses [106]. In cup-ulocytes, NLRP6 was demonstrated to regulate mucus secretion in an inflammatory vesicle and autophagy-dependent manner to prevent invasion by intestinal bacteria [106,107]. Likewise, NLRP6 was found to prevent enterovirus infection, with increased viral load in knockout mice systemically infected with cerebral myocarditis virus or MNV in *Nlrp6* knockout and control mice. NLRP6 was shown to inhibit enterovirus infection through the interferon pathway by binding to viral RNA via the RNA-decapping enzyme DHX15 [108]. Ligands such as double-stranded RNA (dsRNA) produced by many viruses can interact with NLRP6 to induce liquid–liquid phase separation (LLPS). Mutations in a positively charged region of NLRP6 are essential for LLPS, which inhibits dsRNA-induced NLRP6 spot formation, GSDMD cleavage, and cell death, and impairs anti-microbial defenses in mice [107]. Mammalian IECs can sense the virus through MDA-5 and NLRP6 to activate type III IFN signaling or through TLR3 to activate NF-κB signaling [67].

RVs specifically infect host small IECs and have evolved strategies to antagonize the IFN and NF-κB signaling pathways [109]. NLRP9 is specifically expressed in IECs and restricts RV infection [110]. NLRP9b recognizes short double-stranded RNA fragments via the RNA decapping enzyme Dhx9 and interacts with the splice proteins ASC and caspase-1 to form inflammatory vesicle complexes that promote IL-18 and gasdermin D-induced apoptosis [109].

### 3.3. Intestinal Neuroimmunity: How the Gut Neural Network Is Involved in Pathogenic Infections

The ENS controls a variety of intestinal physiological functions such as gut motility, which can be disrupted by infection-induced neuropathy or neuronal cell death [22,23]. Enteric-associated neurons are closely associated with immune cells and continuously monitor and regulate gut homeostatic functions, including motility and nutrient perception [111]. Bacterial intestinal infections can lead to persistent inflammatory changes in the intestine accompanied by a decrease in the number of myenteric neurons caused by NLRP6- and caspase-11-mediated apoptosis. In contrast, intestinal myenteric macrophages can respond rapidly to intestinal infection to limit infection-induced enteric neuronal death through the β2-adrenergic-arginase 1-polyamine axis signaling pathway [111]. Infection by different pathogens of the intestinal tract, including bacteria and parasites, induces a protective phenotype in intestinal myenteric layer macrophages, thus playing a role in protecting enteric neuronal cells and intestinal physiological functions during subsequent infections [112].

The maintenance of tissue homeostasis and injury repair in the gut requires the combined action of multiple cell types, including epithelial cells, immune cells, stromal cells, and glial cells [23]. Glial and pericytes in the intestine are in close contact with vascular endothelial cells and collectively form the gut–vascular barrier (GVB) [113]. Spadoni et al. revealed that GVB coordinates the entry of antigens and bacteria into the bloodstream; intestinal glial cells can transmit and receive signals from enteric neurons and IECs to maintain the GVB, which has a diffusible molecule eight times larger than the blood–brain barrier; murine typhoid *Salmonella* infection enhances the β-catenin signaling pathway in intestinal epidermal cells but inhibits the β-catenin pathway in vascular endothelial cells, leading to bacterial passage through the GVB (Figure 2) [114].

The ENS is the largest neural organ outside the brain and can function largely autonomously in response and adaptation to local challenges [115]. The innervation system in the gastrointestinal tissues is important for the maintenance of normal gastrointestinal function [23]. The gastrointestinal tract is innervated by foreign neurons from the dorsal root ganglion and vagal ganglion and is also controlled by built-in neurons in the muscularis and submucosa [22]. Among them, intestinal injury receptors (Nociceptor Neurons) are the main receptors in the gastrointestinal tract when these disorders of visceral pain and diarrhea occur, triggering inflammatory and pain-protective neural reflexes [116,117]. Injury receptors are capable of interacting with gut microbes and intestinal cells and are involved in the regulation of intestinal mucosal defense. Dorsal root ganglion injury receptors were found to resist colonization, invasion, and intestinal spread of STm. The injury receptor regulates the number of ileal Peyer’s patch (PP) follicle-associated epithelia (FAE) M cells and segmented filamentous bacteria (SFB) by releasing calcitonin gene-related peptide (CGRP), thereby limiting STm invasion [116]. SFB is an intestinal microorganism that colonizes the ileal villi and PP FAE, and SFB induces histone modifications in IECs at retinoic acid receptor-rich motif sites to promote RA-associated defenses against pathogenic bacterial infections to the organism [118]. CGRP is a neuropeptide that regulates neutrophil and dendritic cell responses in the organism [119], regulates M-cell and SFB levels in the intestine to prevent *Salmonella* infection, and plays a key regulatory role in ILC2 homeostasis and response (Figure 2) [120]. These findings reveal an important role of injurious receptor neurons in the perception and defense against intestinal pathogens.

Intestinal mucosal immunity plays a key role in the maintenance of commensal flora and resistance to intestinal bacterial infections [23]. In the past, this role was thought to be mediated mainly by the intestinal immune system and intestinal epithelium, but recent studies pointed out that the ENS may also play an important role [22]. IL-18 has emerged as a key pleiotropic cytokine for homeostasis and host defense in the intestine. In immune cells, IL-18 is a pro-inflammatory cytokine that is required to combat invasive bacterial infections such as STm. However, IL-18 of epithelial and immune cell origin has no protective effect against intestinal STm infection [121]. The enteric nerve directly produces the cytokine IL-18, which in turn increases the expression of AMP in cupped cells to strengthen the intestinal barrier against bacterial invasion [122]. Unlike immune cells and epithelial sources, neurons expressing IL-18 non-redundantly instruct cupped cells to program AMP production, reinforce the sterile intra-mucosal barrier during homeostasis, and help kill intestinal pathogens during infection (Figure 2). The ENS is a key branch of the innate immune response, not only for coordinating mucosal barrier homeostasis but also against invasive bacterial infections [22].

### 3.4. Nutritional Immunity: The Struggle for Micronutrients at the Host–Pathogen Interface

Metal ions are one of the essential nutrients in pathogens and hosts. In order to limit pathogen infection, the host’s innate immune system has evolved a series of mechanisms to limit the use of its own metal ions by pathogenic bacteria, a process known as “nutritional immunity” [123]. Consequently, pathogenic bacteria have also evolved mechanisms to effectively counter host nutritional immunity, usually by utilizing various transport systems to transport metal ions into bacterial cells to satisfy bacterial growth requirements [124]. The ability of pathogenic bacteria to compete with commensal bacteria for nutrients is critical to their survival in the gut [125]. Formerly, attention was focused on how metal limitation inhibits bacterial growth, but there is growing evidence that hosts also employ the toxicity of metals to poison intracellular bacteria directly [126]. In addition to bacteria, many trace elements play a key role in viral infections. For example, zinc inhibits viral protease and polymerase enzymatic cleavage processes, as well as viral attachment, infection, and decapsidation, while playing a regulatory role in T cell-mediated antiviral immunity [127].

Nutritional immunity is an important process that is disrupted during viral–bacterial co-infection. Respiratory syncytial virus (RSV) infection promotes *Pseudomonas aeruginosa* (*P. aeruginosa*) biofilm growth through dysregulation of trophic immunity and further allows apical release of transferrin, thereby increasing iron bioavailability for *P. aeruginosa* biofilm growth in vivo and in vitro [128]. Respiratory viral infections dysregulate host iron homeostasis and promote harmful secondary bacterial infections. This adds to our understanding of the mechanisms of viral–bacterial co-infection. Extracellular vesicles also participate in cross-border trophic transfer during bacterial–viral co-infection [129]. Occupying ecological niches in infection by competing for metal ions is a common modality in co-infection, but studies on nutritional immunity of intestinal pathogens are in their infancy, but we can still mine information from them to gain more insight into the interactions of different pathogens in co-infection.

Iron is involved in DNA replication and cell proliferation, killing viruses by binding lactoferrin and transferrin and interacting with neutrophils. Excessive iron levels may increase viral activity and the mutation rate of viruses [126]. In addition to regulating iron levels in individual cells to control iron-produced cytotoxicity, regulation of iron distribution is also an innate immune mechanism that resists the invasion of pathogens into organisms. The effector of STm, SpvB encoded by pSLT, regulates the hepcidin–ferroportin axis through the IL-6/JAK/STAT3 pathway, leading to a decrease in cellular iron transporters transporting ferritin and exacerbating dysregulation of systemic iron metabolism (Figure 2). At the same time, SpvB promotes the production of pro-inflammatory molecules associated with inflammatory cell infiltration by highly upregulating the TREM-1 signaling pathway [130,131]. A similar mechanism is present in enterovirus infections. Coxsackievirus B3 (CVB3) infection of Balb/c mice induces differential upregulation of gene expression of metallothionein (MT1), divalent metal transporter 1 (DMT1), and hepcidin in the intestine and liver, leading to an increase in the copper/zinc (Cu/Zn) ratio in serum [132]. Reducing free iron levels by adding exogenous lipocalin 2, a host iron chelating protein, reduces the bacterial burden during viral–bacterial co-infection [129].

At the interface between host and pathogen, calmodulin exerts antimicrobial effects by competing for or chelating pathogenic zinc and manganese, inactivating pathogens and weakening their defenses against host immune responses [133]. In some cases, pathogens have developed the ability to compete with calmodulin-mediated intestinal metal starvation. STm overcomes calprotectin-mediated zinc chelation by expressing a high-affinity zinc transporter protein (ZnuACB), allowing STm to proliferate normally in the face of intestinal inflammation and to outcompete commensal bacteria [134]. Most bacteria, such as *E. coli*, *Salmonella*, and *Brucella*, take up zinc and counteract zinc deficiency through the ZnuACB uptake system [135]. Meanwhile, STm utilizes the same strategy to evade the isolation of manganese by calmodulin. The ability of the pathogen to acquire manganese, in turn, promotes the function of SodA and KatN, enzymes that use metals as cofactors to detoxify reactive oxygen species. This manganese-dependent SodA activity allows bacteria to evade calprotectin and reactive oxygen species-mediated neutrophil killing (Figure 2). Thus, manganese acquisition allows STm to overcome host antimicrobial defenses and supports its competitive growth in the intestine [133].

In addition to passive resistance to host nutritional immunity, the virus also takes an active role in promoting infection through the modulation of certain ions. In cell lines and human gut-like organs, RV infects a portion of IECs, and the infected cells release adenosine 5′-diphosphate (ADP), which activates P2Y1 purinergic receptors in surrounding cells leading to increased intracellular calcium ion concentrations that promote the virus’ own replication. It was shown that RV could use the paracrine purinergic signaling pathway of cells to generate intercellular calcium waves, which amplify the dysregulation of IECs and alter the gastrointestinal physiology of infected individuals, triggering diarrhea (Figure 2) [136].

### 3.5. The Gut Microbiota and Co-Infections: A Complex Interaction

The mammalian gut is inhabited by trillions of microorganisms, most of which are bacteria that have co-evolved with their hosts in a symbiotic relationship. A major function of the microbiota is to protect the gut from exogenous pathogens and potentially harmful endogenous microorganisms through several mechanisms, including direct competition for limited nutrients and modulation of the host’s immune response [137]. The main contributions of the microbiota to the host include carbohydrate digestion and delivery, vitamin production, development of gut-associated lymphoid tissues, the polarization of the gut-specific immune response, and prevention of pathogen colonization. In turn, the gut immune response induced by commensal flora regulates the composition of the microbiota. However, the reciprocal relationship between the host and microbiota is disrupted when pathogens invade. The structure as well as the abundance of the intestinal microbiome is altered, which can cause inhibitory or facilitative effects on the pathogen [137]. Understanding the interactions of bacterial–viral co-infection in the gut microbiome can help develop strategies to manipulate the microbiome to fight infectious diseases [125].

Intestinal pathogen co-infection influences changes in microbial abundance associated with single infections. Shilu Mathew et al. evaluated the composition of the intestinal microbiota in children infected with two major viruses, RV and NoV, and two pathogenic bacteria, EAEC and EPEC, alone or in co-infections. The results showed that the abundance of the Bifidobacterium family as probiotics increased with the severity of the mixed viral–bacterial infections. The relative numbers of Bifidobacteria were significantly reduced in RV and NoV infections and more so in co-infected pathogenic *E. coli*. Although mixed EAEC infections resulted in significant microbiota differences compared to viral infections alone or mixed EPEC infections, co-infections of both pathogenic *E. coli* exacerbated clinical symptoms in children [13].

The intestinal microbiota affects virus replication in the gastrointestinal tract and systemic transmission. Sharon K. Kuss et al. cleared the intestinal flora of mice by antibiotics and subsequently infected them with PV. It was found that mortality was higher in untreated mice than in antibiotic-treated mice and that reintroduction of fecal bacteria into the antibiotic-treated mouse group enhanced PV disease. These results suggest that mice containing microbiota support PV replication more effectively than mice lacking microbiota (Figure 2) [55]. PV binding to specific microbial-associated surface polysaccharides enhances viral heat stability and attachment to host cells. The virus may have evolved to use gut microbes at the most suitable sites for transmission as triggers for replication [52,56]. In the same vein, antibiotics prevented persistent MNoV infection, an effect that was reversed by the replenishment of the bacterial microbiota. Antibiotics did not prevent tissue infection or affect systemic virus replication but were particularly effective in the intestine. The antiviral cytokines interferon λ receptor, Ifnlr1, and the transcription factors Stat1 and Irf3 are required for antibiotics to prevent viral persistence. Thus, the bacterial microbiota promotes enterovirus persistence in a manner that is counteracted by specific components of the innate immune system [49].

The enterovirus group also has an important role in the maintenance of intestinal homeostasis. Supplementation with specific viruses can help immunodeficient mice to improve their ability to contract enterovirus infection, and this resistance to infection can be transferred between hosts through the lateral transmission of the commensal flora. The results highlight the role of enteroviruses in resistance to intestinal infections and help to investigate the mechanisms of enterovirus group–host interactions [41]. NoV infection protects the intestine from injury by eliciting an IFN-I response in the mouse colon, where IFN-I acts on IECs to increase the proportion of CCR2-dependent macrophages and IL-22-producing natural lymphocytes, thereby promoting pSTAT3 signaling in IECs. MNV infection plays a protective role in DSS-induced intestinal injury and intestinal infection caused by *Citrobacter rhamnosus* (Figure 2) [42,138].

## 4. Conclusions

It was demonstrated that bacterial and viral co-infections could occur in multiple sites, including the respiratory tract and the intestinal tract. The intestinal tract is an intricate and complex environment, which is further complicated by pathogenic stimuli and comprises various aspects. Emerging technologies such as high-throughput sequencing (metagenomics, metabolomics, metallomics, etc.) allow us to understand intestinal pathogen infections from a comprehensive perspective; organoid models simulate the intestinal environment more realistically than cells alone in infection assays, and single-cell sequencing provides a thorough understanding of intracellular development from a cellular perspective. In conclusion, the study of intestinal pathogen co-infection is entering a new and more comprehensive phase, but there is more work to be conducted to elucidate further the mechanisms of bacterial–viral–gut interactions. By focusing on pathogen–pathogen and pathogen–host interactions, rather than just on the pathogen, it is possible to leverage these pathways to identify new therapeutic targets.

## Figures and Tables

**Figure 1 ijms-23-02311-f001:**
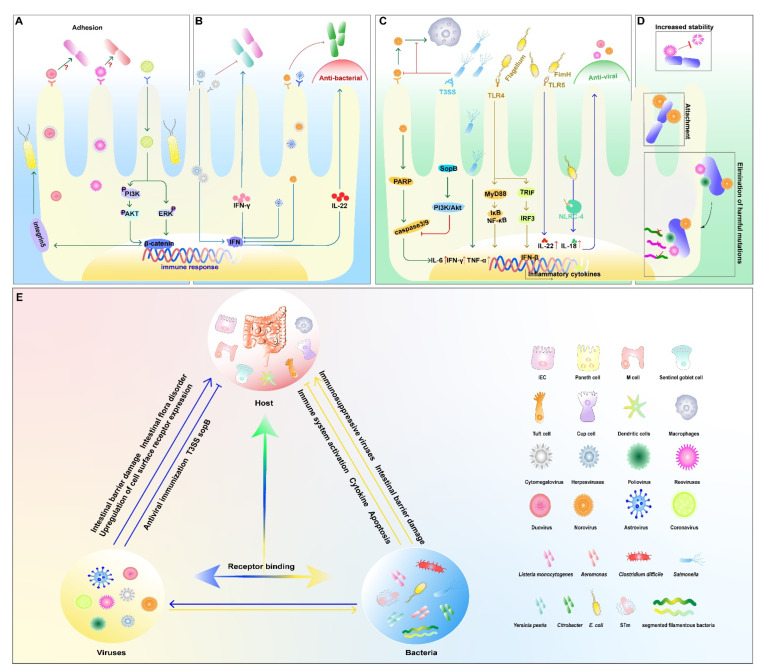
Bacterial and viral co-infection in the intestine. (**A**) viruses promote bacterial infections; (**B**) viruses inhibit bacterial infections; (**C**) bacteria inhibit viral infections; (**D**) bacteria promote viral infections; (**E**) the relationship among the host, viruses, and bacteria. An arrow between two elements means that the side being pointed at is activated. Conversely, the symbol (├) represents inhibiting effect. The symbol (↑) following the proteins represents increased expression.

**Figure 2 ijms-23-02311-f002:**
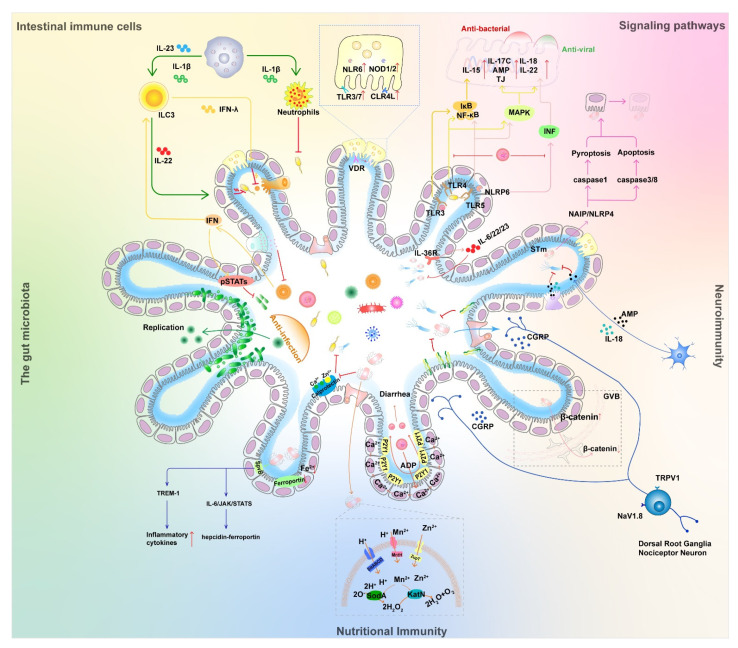
Competition scenario of bacterial and viral co-infection and their effect on host immunity. This figure broadly classifies interactions among host, viruses, and bacteria and describes the details in five aspects: intestinal cells and their secreted effectors, signaling, neuroimmunity, nutritional immunity, and the gut microbiome. An arrow between two elements means that the side being pointed at is activated. Conversely, the symbol (├) represents an inhibiting effect. The symbol (↑) following the proteins represent increased expression, and the symbol (↓) represents decreased expression. See Figure 1 for illustrations of various cells and pathogens.

## Data Availability

All datasets generated for this study are included in the article.

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
