# Peer review of "Bacterial and Viral Co-Infection in the Intestine: Competition Scenario and Their Effect on Host Immunity"

_ijms, 2022, doi:10.3390/ijms23042311_

Round 1

Reviewer 1 Report

The authors describe the “Bacterial and Viral Co-infection in the Intestine: Competition Scenario and Their Effect on Host Immunity”. In this review, they discuss the different scenarios triggered by different orders of bacterial and viral infections in the gut and then summarize the possible mechanisms of antagonism involved in their co-infection. The authors also explore the regulatory mechanisms of bacterial-viral co-infection at the host intestinal immune interface from multiple perspectives. I believe the review is very well written, well organized and it will open new research thinking in this field of science.

Author Response

Reviewer 1: The authors describe the “Bacterial and Viral Co-infection in the Intestine: Competition Scenario and Their Effect on Host Immunity”. In this review, they discuss the different scenarios triggered by different orders of bacterial and viral infections in the gut and then summarize the possible mechanisms of antagonism involved in their co-infection. The authors also explore the regulatory mechanisms of bacterial-viral co-infection at the host intestinal immune interface from multiple perspectives. I believe the review is very well written, well organized and it will open new research thinking in this field of science.

--Thank you for your approval and support of this article. The attachment is the latest manuscript. Thank you again.

Reviewer 2 Report

Summary:

The authors review examples of bacterial and viral coinfection with emphasis on order of infection and microbial combination. They give evidence that refutes the simplistic model that pathogen quantity is directly proportional to infection severity, and promote the model that like any ecological system, the gut microbiome is complicated.

Comments:

This review is generally thorough and well-written. A few minor errors have slipped through the copy-editing process, those I saw are listed below. The authors may wish to consider those revisions.

Line 33, animals also lose: animals lose. As written, the "also" in this sentence suggests humans (and in context of the previous sentence, children) lose their economic value with diarrheal disease. Changing "humans, various" on this line to "human disease, various" would help focus economic value on the livestock only.

Line 38, one simultaneous: single-pathogen. Simultaneous indicates two infections at the same time, which is the definition of co-infection, unless the authors wish to clarify this sentence in another way.

Lines 44 and elsewhere, et al.: et al. is italicized

Lines 51–52, All co-infections...: This is the definition of co-infection and is unnecessary, please consider removing.

Figure 1A: question marks are overlapping arrows in figure, please move down for clarity.

Lines 168–175: It would be nice if this paragraph connected to effects of immune exhaustion or age-related reductions in immune responsiveness that might be cogent to the example.

Line 206, in vitro: in vitro is italicized.

Line 232, poliovirus: PV. Inconsistent use of abbreviation.

Line 277, The gut is large: The gut is a large.

Figure 2: Some font is too small on the labels, e.g. Ferroportin, SpoB, and P2Y1.

Author Response

Reviewer 2:

The authors review examples of bacterial and viral coinfection with emphasis on order of infection and microbial combination. They give evidence that refutes the simplistic model that pathogen quantity is directly proportional to infection severity, and promote the model that like any ecological system, the gut microbiome is complicated.

This review is generally thorough and well-written. A few minor errors have slipped through the copy-editing process, those I saw are listed below. The authors may wish to consider those revisions.

- Fixed as requested. We have carefully revised our paper accordingly.

Specific comments:

Line 33, animals also lose: animals lose. As written, the "also" in this sentence suggests humans (and in context of the previous sentence, children) lose their economic value with diarrheal disease. Changing "humans, various" on this line to "human disease, various" would help focus economic value on the livestock only.

- Fixed as requested.

Line 38, one simultaneous: single-pathogen. Simultaneous indicates two infections at the same time, which is the definition of co-infection, unless the authors wish to clarify this sentence in another way.

- Fixed as requested.

Lines 44 and elsewhere, et al.: et al. is italicized

- Fixed as requested.

Lines 51–52, All co-infections...: This is the definition of co-infection and is unnecessary, please consider removing.

- Fixed as requested. Figure 1A: question marks are overlapping arrows in figure, please move down for clarity.

-Fixed as request. Please see the new Figure 1 in the revised manuscript.

Lines 168–175: It would be nice if this paragraph connected to effects of immune exhaustion or age-related reductions in immune responsiveness that might be cogent to the example.

- Thank you for your suggestion, and sorry for we cannot modify this section at this time. Yes, this paragraph is related to immune failure and age-induced immune decline, but the detailed mechanism is complicated and not elucidated yet. We think it is really not easy to articulate the main point in short paragraphs.

Line 206, in vitro: in vitro is italicized.

- Fixed as requested.

Line 232, poliovirus: PV. Inconsistent use of abbreviation.

- Fixed as requested.

Line 277, The gut is large: The gut is a large.

- Fixed as requested.

Figure 2: Some font is too small on the labels, e.g. Ferroportin, SpoB, and P2Y1.

- Fixed as requested. Please see the new Figure 2 in the revised manuscript.

Again, I would thank you for your support and help!

Sincerely yours,

Pengpeng Xia

Ph.D, Associate Professor

Yangzhou University College of Veterinary Medicine

Yangzhou, 225009, China

Tel.: (0086)-514-87979033

Fax: (0086)-514-87311374
